# Quantitative Assessment for the Spatiotemporal Changes of Ecosystem Services, Tradeoff–Synergy Relationships and Drivers in the Semi-Arid Regions of China

**Yongge Li [1,2], Wei Liu [1,\*], Qi Feng [1], Meng Zhu [1], Linshan Yang [1] and Jutao Zhang [1]**

[1] Key Laboratory of Ecohydrology of Inland River Basin, Northwest Institute of Eco-Environment and Resources, Chinese Academy of Sciences, Lanzhou 730000, China; liyongge@lzb.ac.cn (Y.L.); qifeng@lzb.ac.cn (Q.F.); zhumeng@lzb.ac.cn (M.Z.); yanglsh08@lzb.ac.cn (L.Y.); jutzhang@lzb.ac.cn (J.Z.)

[2] University of Chinese Academy of Sciences, Beijing 100049, China

\* Correspondence: weiliu@lzb.ac.cn

**Abstract:** Ecosystem services in arid inland regions are significantly affected by climate change and land use/land cover change associated with agricultural activity. However, the dynamics and relationships of ecosystem services affected by natural and anthropogenic drivers in inland regions are still less understood. In this study, the spatiotemporal patterns of ecosystem services in the Hexi Region were quantified based on multiple high-resolution datasets, the InVEST model and the Revised Wind Erosion Equation (RWEQ) model. In addition, the trade-offs and synergistic relationships among multiple ecosystem services were also explored by Pearson correlation analysis and bivariate spatial autocorrelation, and redundancy analysis (RDA) was also employed to determine the environmental drivers of these services and interactions. The results showed that most ecosystem services had a similar spatial distribution pattern with an increasing trend from northwest to southeast. Over the past 40 years, ecosystem services in the Hexi Region have improved significantly, with the water retention and soil retention increasing by $87.17 \times 10^8$ m³ and $287.84 \times 10^8$ t, respectively, and the sand fixation decreasing by $369.17 \times 10^4$ t. Among these ecosystem services, strong synergistic relationships were detected, while the trade-offs were found to be weak, and showed significant spatial heterogeneity in the Hexi Region. The spatial synergies and trade-offs in the Qilian Mountains were 1.02 and 1.37 times higher than those in the Hexi Corridor, respectively. Human activities were found to exacerbate the trade-offs between ecosystem services by increasing water consumption in the Hexi Corridor, with the exception of carbon storage. In particular, there were significant tradeoffs between food production and water retention, and between soil retention and habitat quality in the oases of the Hexi Corridor, which is affected by rapid population growth and cropland expansion. Additionally, precipitation, temperature and vegetation cover in the Qilian Mountains have increased significantly over the past four decades, and these increases significantly contributed to the enhancements in water retention, carbon storage, habitat quality, soil retention and food production. Nevertheless, the amount of sand fixation significantly decreased, and this was probably associated with the reduction in wind speed over the past four decades. Our results highlighted the importance of climate wetting and water resource management in the enhancement of ecosystem services and the mitigation of food production trade-offs for arid inland regions.

**Keywords:** ecosystem services; water retention; sand fixation; trade-offs; Hexi corridor; Qilian mountains

## 1. Introduction

Ecosystem services refer to all of the benefits and contributions that humans derive from natural systems [1], including provisioning, regulating, supporting and cultural services; they have been crucial for human wellbeing and sustainability at different scales [2,3]. About 60% of global ecosystem services are being degraded, directly threatening regional and global ecological security, economic development, and people's livelihoods and health due to global changes in climate and land use/cover, or through direct and indirect interactions among ecosystem services [3–6]. In order to meet the growing human demands on natural resources, and to maintain the essential ecosystem functions and resilience, we should have a deeper understanding of the complex relationships between ecosystem services and drivers, which ultimately provides the base for the implementation of sustainable management strategies [7,8].

However, the difficulty in assessing the linkages and relationships of ecosystem services have brought important challenges to policymakers, environmental planners and researchers [9,10]. Relationships among ecosystem services occur when multiple services respond to the same drivers of change, or when interactions among services themselves cause a change in one service that alters another [4]. Specifically, ecosystem service trade-offs, a competitive relationship, occur when one ecosystem service is enhanced at the cost of the reduction of another service; synergies occur when multiple services increase or decrease simultaneously [11,12]. Meanwhile, positive (synergistic) and negative (trade-off) relationships among ecosystem services are substantially influenced by policy interventions and environmental variability [13]. Furthermore, the intensification of trade-offs between ecosystem services has increased, and has experienced rapid changes in global and certain regions, which exacerbate the vulnerability in these regions [12,14,15]. For example, excessive afforestation programs in drylands aggravated the tradeoff of water yield and soil conservation, and threatened ecosystem sustainability [15].

The assessment of the spatial distribution of ecosystem services plays vital roles in the identification of trade-offs and synergies among ecosystem services for decision support. Currently, more systematic and spatially oriented tools for ecosystem service assessment and trade-off analysis are available, including the Integrated Valuation of Ecosystem Services and Tradeoffs (InVEST), Artificial Intelligence for Ecosystem Services (ARIES), Land Utilisation Capability Indicator (LUCI), and Multiscale Integrated Models of Ecosystem Services (MIMES) [16–20]. Compared with other models, the InVEST model has been amenable to extensive use, as it can be independently tested and open sourced [15,21–23]. In addition, there are many methods to assess and account for trade-offs and synergies among ecosystem services, mainly including correlation analysis, redundancy analysis (RDA), overlap analysis (e.g., the local indicators of spatial association), regression analysis and scenario analysis [24,25]. For example, based on different climate scenarios, Xu et al. [26] identified geographic factors that influence ecosystem service relationships in the Belt and Road region from 2010 to 2030. Renard et al. [27] showed that the spatial distribution of ecosystem services, based on redundancy analysis, was related to biophysical and socioeconomic drivers in Quebec, Canada. A recent study quantified the relationship between forest cover and indicators associated with poverty in Brazil, using the local indicators of spatial association (LISA) analysis [28]. Among these methods, the widely utilized methods were correlation analysis by correlation coefficients and overlap analysis based on spatial association, which improved our knowledge of the quantitative and spatial dependence in the relationships of ecosystem services [29–32].

Ecosystem services have been broadly recognized and studied for the improvement of biodiversity conservation and human wellbeing across the globe (e.g., Europe [33,34], Canada [27], the United States [35,36], Australia [37], Brazil [38], China [39]), with a notable few in arid inland regions (e.g., Central Asia, North Africa), the Qinghai-Tibetan Plateau [40,41], or polar regions [2]. The arid inland regions are an important component of the dryland ecosystems that comprise about 41% of the earth's land surface and support more than 38% of its population [42]. However, these regions are considered fragile and

sensitive to desertification due to climate change and inappropriate human activities. As a result of the limited data availability resulting from harsh climates and complex landscapes, the dynamics and relationships of ecosystem services, as affected by changes in climate and land use in inland regions, are still less understood. As a typical inland region in Northwest China, the Hexi Region, a mountain-oasis-desert ecosystem, is primarily characterized by a rapidly warming and wetting climate in the mountains of the region (i.e., the Qilian Mountains), and extensively developed irrigated agriculture and dense population in the plains (i.e., the Hexi Corridor). Especially, the glaciers, permafrost and perennial snow in the Qilian Mountains in the arid region of western China are important water conservation areas, which mainly act as the sources of many large inland rivers over the Hexi Region, including the Shiyang River Basin [43,44], the Heihe River Basin [45], and the Qinghai Lake Basin [46,47]. The Qilian Mountains are also characterized by rich biodiversity and a high value of ecosystem productivity due to adequate rainfall in comparison with the Hexi Corridor. Sandy land, bare land and grassland were the predominant land cover types in the Hexi Region, occupying 30.33%, 17.20% and 31.42% of the total area in 2018. The ecosystems in the Hexi Region provide a wide range of important services, including food production, biodiversity conservation, carbon storage, water yield, water retention and soil conservation [31,43,44], which support ecological ecosystem stability, and social and economic development in the Hexi Region. The dominant ecosystem services and ecological functions in the Hexi Region are water retention, sand fixation (sandstorm prevention), soil retention and biodiversity conservation [48], and a better understanding to the spatial and temporal heterogeneity of these services and their drivers is vital to sustainable ecosystem management [49].

Over the past decades, dramatic climate change and anthropogenic activities have resulted in great challenges to the Hexi Region in terms of ecosystem functions and services. Glacier retreat [50], grassland degradation, desertification and water retention capacity have declined [46], significantly threatening the ecological security in and nearby the Hexi Region [51]. The Hexi Region is adjacent to deserts coupled with relatively low vegetation coverage, low rainfall, droughts, and frequent strong winds [52]; wind erosion also has been affecting the ecological security in Hexi Region. The evaluation of sand fixation services is important for the reduction of the hazard of wind erosion [53]. Furthermore, the Hexi Region has undergone remarkable changes over the recent decades due to rapid urbanization and cropland expansion, and a series of ecological restoration initiatives. Thus, an evaluation of the ecosystem service dynamics and drivers will enable us to better understand the evolution mechanism of ecosystem services in arid inland regions. Although there are scattered case studies in the Hexi Region (e.g., a small watershed, city or local areas) [44–46] concerning food production, habitat quality, carbon storage, water yield, water retention and soil conservation, few of them have considered sand fixation services, and systematical and comprehensive assessments of multiple ecosystem services as affected by climate change and human activities remain few. There is an urgent need to study the spatiotemporal changes of multiple ecosystem services, including sand fixation services, trade-offs and synergies, and drivers for the improvement of and the ensuring effective management of ecosystem services and ecological security, and the enhancement of human wellbeing in arid inland regions.

Remote sensing is regarded as an important tool which could effectively enable spatially explicit estimates of ecosystem services to be made [54,55]. It provides various spectral information from different land cover types, which could be further used to assess the spatiotemporal changes of multiple ecosystem services at varying spatial and temporal scales or resolutions [56]. Taking remote sensing-based data as inputs into the InVEST model and the Revised Wind Erosion Equation (RWEQ) model—including the land cover types, vegetation indexes, evapotranspiration, precipitation, temperature, soil properties and topography—could significantly increase the quality of evaluation results [53,57]. In particular, land cover and NDVI were the predominant variables used in the evaluation

of different ecosystem services [58]. In addition, soil and vegetation carbon datasets derived from remote sensing and machine learning algorithms were also applied in the InVEST model, which enables more spatially explicit estimates for carbon sequestration to be made. As a result, the quality of ecosystem service evaluation results was primarily affected by the accuracy of raw data, which could be improved through the introduction of more high-quality remote sensing data to the models. Many previous studies have demonstrated that the LUCC products used to evaluate ecosystem services in the Hexi region were generally characterized by coarser categories (e.g., croplands, forests and grassland) [44,46], as many fundamental parameters for finer land-cover category over spatial extents were hard to quantify with limited literature results or other available datasets. Hence, higher-quality LUCC data with finer classifications and other remote-sensing based key variables should be employed in the evaluation of the ecosystem services at a regional scale.

Therefore, the primary aims of this study were as follows: (1) to quantify the spatio-temporal changes of ecosystem services in both the mountains and plains of the Hexi Region based on the InVEST model and the RWEQ model, (2) to investigate trade-offs and synergies among ecosystem services in the Hexi Region based on Pearson correlation analysis and the bivariate Local Moran's I analysis, and (3) to identify the environmental drivers of ecosystem services by RAD analysis.

## 2. Materials and Methods

### 2.1. Study Area

The Hexi Region (92.29°–104.76°E, 35.75°–42.78°N) is located in the north edge of the Qinghai-Tibetan Plateau, with a total area of $3.8 \times 10^5$ km², including the Hexi Corridor (660–2000 m a.s.l.) and Qilian Mountains (2000–5838 m a.s.l.) (Figure 1). The Hexi Region is characterized by typical arid and semi-arid climates, with a mean annual precipitation, potential evapotranspiration and temperature ranging from 50 to 700 mm, 600 to 1600 mm and −12 to 12 °C, respectively. The climate tends to be colder and wetter as the elevation increases, and precipitation shows an overall decreasing trend from the southeast to the northwest part of the region. The Qilian Mountains are widely covered with modern glaciers at elevations greater than 4500 m, which are source areas of many inland rivers, providing valuable water resources for oases in the Hexi Corridor [52]. As affected by topography, the landscapes in the Hexi Region are characterized by obvious vertical zonality. The major landscapes along the elevation gradient are temperate desert/oases, desert steppe, montane shrub steppe, montane forest steppe, subalpine shrub meadow, alpine meadow/steppe, alpine desert, and permanent snow/glaciers. In addition, the predominant land cover types in the Hexi Region are sandy land, bare land and grassland, suggesting a fragile environment in the arid inland regions (Figure 2). Moreover, the forest and high- and medium-cover grassland are mainly located in the middle-eastern part of Qilian Mountains, and the cropland is primarily distributed in the Huangshui Valley on the eastern side of Qinghai Lake and the oases of the Hexi Corridor.

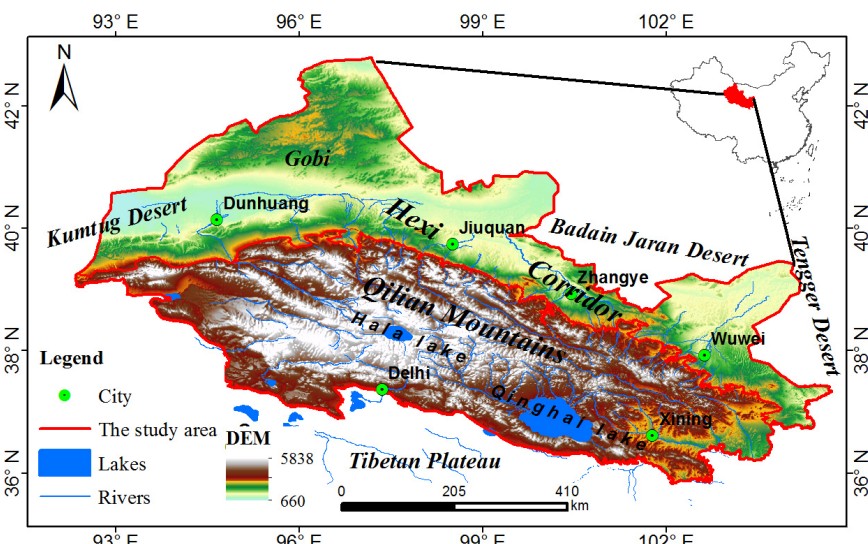

**Figure 1.** Geographical location of the Hexi Region.

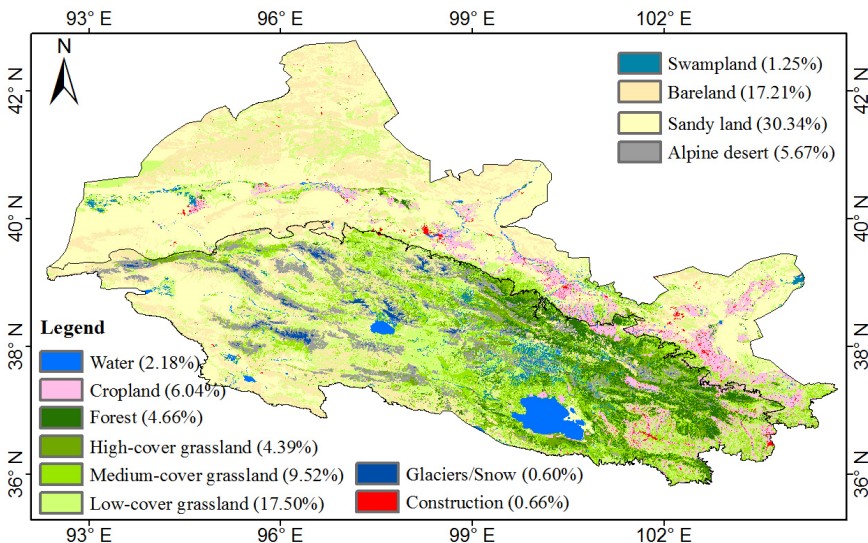

**Figure 2.** The spatial distribution of LUCC from 2018 in the Hexi Region.

*2.2. Data Sources*

In order to evaluate the patterns, relationships and drivers of six ecosystem services, we need to consider and apply multiple datasets, such as land use and land cover change (LUCC) data, and other meteorological, pedological, remote sensing and socioeconomic data. The details are as follows.

2.2.1. LUCC Datasets

The LUCC datasets (1980–2018) were from the National Land-Use/Cover Database of China (NLUD-C), obtained from the Data Center for Resources and Environmental Sciences, Chinese Academy of Sciences (RESDC) (http://www.resdc.cn, accessed on 23 June 2021). The NLUD-C datasets were generated by human–machine interactive interpretation based on Landsat imagery. Using random sampling and field surveys for comparison, the overall classification accuracies exceeded 90% [51,59,60]. The NLUD-C datasets divided the land-use categories into 25 land-use types in total, with fewer subclasses for forests and more classes for arable land and water. In order to reflect the complex landscapes patterns in the Hexi Region, the original 25 land-use types were divided into 13

major categories: (1) cropland; (2) forest; (3)shrubland (4) high-cover grassland; (5) medium-cover grassland; (6) low-cover grassland; (7) water, including rivers, reservoirs and ponds; (8) glaciers or snow; (9) construction land, (10) sandy land; (11) swampland; (12) bare land; and (13) alpine desert with an average elevation of 4300 m. It should be noted that grasslands with vegetation cover >50%, 20–50%, and 5–20% are classified as the high-, medium-, and low-cover grasslands, respectively.

### 2.2.2. Meteorological and Remote Sensing Data

The temperature and precipitation data included both annual gridded datasets at 1 km resolution [61] and daily station observations (1980–2018) (including the daily wind speed, daily temperature, daily precipitation and daily solar radiation, etc.) from the Meteorological Data Service Centre (CMDC) (http://data.cma.cn/, accessed on 23 June 2021). Potential evapotranspiration ($ET_0$) is an essential component of both climate and hydrology cycles, and significantly influences vegetation growth and water consumption. It can be calculated by the modified Penman–Monteith equation with meteorological data [62,63]. Long-term series of the daily snow depth dataset in China (1979–2020) derived from passive microwave remote sensing data [64,65] and the Landsat-based continuous monthly 30 m × 30 m land surface NDVI dataset in Qilian Mountain (1986–2018) were obtained from the National Tibetan Plateau Data Center (http://data.tpdc.ac.cn, accessed on 23 June 2021). The maximum value composition (MVC) method was used to synthesize the monthly NDVI products on the surface using the reflectivity data of Landsat 5 and Landsat 8 [66,67]. The food production and sand fixation services were calculated by the NDVI dataset.

### 2.2.3. Soil and DEM Data

The soil datasets (including soil depth, bulk density, clay content, silt content, sand content, and soil organic carbon content) were downloaded from the SoilGrids products at a 250-m resolution (https://soilgrids.org/, accessed on 23 June 2021), having been released by the ISRIC (International Soil Reference Information Centre)—World Soil Information [68]. The carbon density in aboveground biomass, belowground biomass, soil, and dead organic matter was mainly derived from existing literature and research [43,69–72]. The Digital Elevation Model (30 m) was obtained from the Geospatial Data Cloud (http://www. gscloud.cn, accessed on 23 June 2021), and was used to calculate aspect and slope.

### 2.2.4. Socioeconomic Data

The statistical information on the main grain yield (including wheat, corn and tubers) and three types of livestock products as meat (pork, beef, and mutton), milk and poultry eggs were taken from the Rural Statistical Yearbook in Gansu Province (http://tjj.gansu.gov.cn/, accessed on 23 June 2021) and Qinghai Province (http://tjj.qinghai.gov.cn/, accessed on 23 June 2021). The traffic network and rivers network data were acquired from the 1:1 million National Basic Geographic Database (https://mulu.tianditu.gov.cn/, accessed on 25 June 2021). The gross domestic product (GDP) and population (POP) gridded dataset at 1 km resolution originated from the Data registration and publishing system of Resources and Environment Science Data Center of the Chinese Academy of Sciences (http://www.resdc.cn/, accessed on 25 June 2021).

### 2.2.5. Environmental Variables

Ecosystem services are generally driven by both natural and anthropogenic factors. In this study, in order to explore the drivers of ecosystem services, we selected 15 environmental variables according to the previous literature, including: eleven natural factors, i.e., elevation (ELE), aspect (ASP), slope (SLPE), potential evapotranspiration (ET0), mean annual temperature (MAT), mean annual precipitation (MAP), mean annual wind speed

(MAW), fractional vegetation cover (FVC), the rainfall erosivity index (R), Shannon's diversity index (SHDI), and river network density (RRD), and four social and economic factors, i.e., road density (ROD), population size (POP), gross domestic product (GDP), and cropland area (CROP).

*2.3. Methods*

The flowchart in Figure 3 summarizes the overall flow of the research.

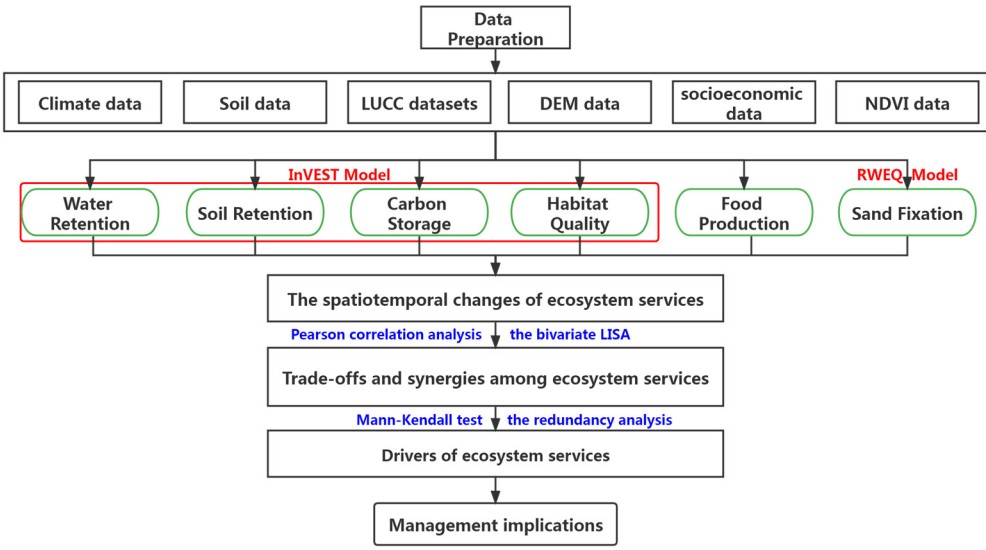

**Figure 3.** Analytical framework applied in the present study.

The InVEST software is a suite of models which was designed for the spatial mapping and valuing of ecosystem services from terrestrial, freshwater, marine, and coastal ecosystems [73]. The InVEST model is amenable to extensive use, as it is open sourced and can be independently tested [15,21–23]. Carbon storage, water yield, soil retention, and habitat quality in the Hexi Region were evaluated using InVEST software (Version.3.9.2), and were associated with LUCC change. We also used the Revised Wind Erosion Equation (RWEQ) model to quantitatively estimate the sand fixation service based on wind erosion and sediment transport by wind [74]; we comprehensively considered the influence of wind speed, precipitation, temperature, soil texture, topography, and vegetation cover on wind erosion [53]. The details are as follows.

2.3.1. Water Retention

The Qilian Mountains are an important water conservation area in the Hexi region, and are also a priority area for biodiversity conservation in China. Water retention (WR) reflects the integrated effect of vegetation, water bodies and soil. Meanwhile, it is characterized by complex processes in which vegetation redistributes rainfall through the canopy, understory, litter and soil layers. This study first used the water yield module of the InVEST model to estimate the water yield based on water balance. Then, on the basis of water yield, the water retention was calculated by the topography, soil thickness and permeability, which aimed to correct the water yield [75]. The formula for calculation is as follows [31,75]:

$$WR = min(1, 0.3TI) \times min(1, K_{sat}/300) \times min(\frac{}{V}, 1) \times Y_x \tag{1}$$

$$TI = \lg[D_a/Soil_d \times P_{sl}] \tag{2}$$

where *WR* is the annual water retention capacity (mm), *TI* is the topographic index (dimensionless) obtained from DEM, $K_{sat}$ is the soil saturated hydraulic conductivity (cm·d⁻¹) calculated by the soil texture and soil bulk density [76], *V* is the velocity coefficient (dimensionless), and $Y_x$ is the annual water yield (mm) calculated by the Water Yield model of the InVEST model. $D_a$ is the number of grids in the catchment area, $Soil_d$ is the soil depth (mm), and $P_{sl}$ is the slope ratio. The water yield model is based on a simple water balance assuming that all of the water in excess of the evaporative loss arrives at the outlet of the watershed [73]; the water yield formula is expressed as [77]:

$$Y_x = (1 - \left(\frac{AET_x}{P_x}\right)) \times P_x \tag{3}$$

$$\frac{AET_x}{P_x} = 1 + \frac{PET_x}{P_x} - [1 + \left(\frac{PET_x}{P_x}\right)^{w_x}]^{1/w_x} \tag{4}$$

$$w_x = Z \times \frac{AWC_x}{P_x} + 1.25 \tag{5}$$

$$AWC_x = \min(D_{soil}, D_{root}) \times PAWC_x \tag{6}$$

where $AET_x$ is the annual actual evapotranspiration (mm); $P_x$ is the annual precipitation amount(mm); $PET_x$ is the potential evapotranspiration calculated by the Penman–Monteith equation [62]; $\omega(x)$ is a nonphysical parameter that characterizes the natural climatic-soil properties; *Z* is the Zhang parameter, which is an empirical constant; $AWC_x$ is the volumetric plant available water content (mm); $D_{soil}$ is the root-restricting layer depth (mm); $D_{root}$ is the vegetation root depth (mm); and $PAWC_x$ is the plant available water capacity, which can be calculated indirectly through the soil texture [78].

### 2.3.2. Soil Retention

The primary cause of accelerated soil erosion is attributed to human activities and related land use change [79]. Based on sheetflow erosion, the soil retention (SR) is calculated by considering the land use type, climate, soil and topography, as follows [26]:

$$SR = R \times K \times LS \times (1 - C \times P) \tag{7}$$

where *SR* is the amount of soil retention, which is calculated by subtracting the actual soil erosion in the current land use type and management from the potential soil erosion in bare soil (t· hm⁻² · yr⁻¹); *R* is the rainfall erosivity (MJ·mm (hm² · hr · yr)⁻¹), calculated by the annual precipitation; *K* is the soil erodibility(t· hm² ·hr (MJ·hm²·mm)⁻¹), calculated using the EPIC model, based on the sand, silty sand, clay and the organic carbon content [80]; *LS* is the slope-length gradient factor(dimensionless); *C* is the crop-management factor (dimensionless) [79]; and *P* is the support practice factor.

### 2.3.3. Carbon Storage

InVEST's Carbon Storage and Sequestration model aggregates the amount of carbon stored in different carbon pools according to land use datasets and carbon density data of various land types. The model assumes that carbon storage changes over time are due to land use/cover conversion from one type to another [73]. The main equations of the model are as follows [81]:

$$C_{tot} = C_{abo} + C_{bel} + C_{soil} + C_{dead} \tag{8}$$

where $C_{tot}$ refers to the total amount of carbon storage (t·hm⁻²), and $C_{abo}$, $C_{bel}$, $C_{soil}$, and $C_{dead}$ refer to the amount of carbon storage in aboveground biomass, belowground biomass, soil, and dead organic matter, respectively [19].

### 2.3.4. Habitat Quality

Biodiversity has been closely linked to the production of ecosystem services. To some extent, the habitat quality represents the biodiversity of a landscape, estimating the quality of the habitat and degradation by analyzing maps of land use and land cover in conjunction with threats to biodiversity; all of the threats on the landscape are additive [73]. Habitat quality can be calculated with the InVEST model using the following equation [26]:

$$Q_{xj} = H_j \times \left[1 - \left(\frac{D_{xj}^z}{D_{xj}^z + k^z}\right)\right] \tag{9}$$

$$D_{xj} = \sum_{r=1}^{R} \sum_{y=1}^{y_r} \left(\frac{w_r}{\sum_{r=1}^{R} w_r}\right) r_y i_{rxy} \beta_x S_{jr} \tag{10}$$

where $Q_{xj}$ is the habitat quality in grid cell $x$ with land use and land cover type $j$, $D_{xj}$ is the total threat level in the grid cell $x$ with LULC type $j$, z is a constant that equals 2.5, $k$ is the half-saturation constant, and $H_j$ is the habitat suitability of LULC type $j$. $R$ is the number of threat factors; $y$ indexes all of the grid cells on threat r's raster map; $Y_r$ indicates the set of grid cells on threat r's raster map; $w_r$ is the weight of the threat factor r, with a value between 0 and 1; $r_y$ is the threat factor value of grid y; $i_{rxy}$ is the impact of threat r that originates in grid cell $y$; $\beta_x$ is the level of accessibility in grid cell $x$, where 1 indicates complete accessibility; and $S_{jr}$ indicates the sensitivity of LULC type j to threat factor $r$, where values closer to 1 indicate greater sensitivity [82]. In this study, we took cropland, construction land, road and bare land as threat sources.

### 2.3.5. Food Production

Ecosystem services support human life in many ways, and food production is essential for human supply [83]. Existing studies have shown that there is a significantly positive relationship between food production and NDVI, and food production varies with different land cover types [84,85]. In this study, we mainly consider three main grains, including crops of wheat, corn and tubers, and three types of livestock products, including meat (pork, beef, and mutton), milk and poultry eggs. According to the normalized deference vegetation index and statistical yearbook data, we calculated the food production of cropland and grassland in the Hexi Region using the following formula [85]:

$$FP_x = NDVI_{x,j}/NDVI_{sumj} \times S_{sumj} \tag{11}$$

where $FP_x$ represents the food production on grid $x$, $NDVI_x$ represents the normalized vegetation index on grid $x$, $NDVI_{sumj}$ is the sum of the normalized vegetation index values of land cover type $j$, and $S_{sumj}$ is the total output of agricultural products corresponding to each land cover type $j$.

### 2.3.6. Sand Fixation

Wind erosion is an important factor affecting the ecological security in the Hexi Region. Sand fixation is also known as sand prevention, which refers to the sand retained in an ecosystem within a certain period [39]. We used the Revised Wind Erosion Equation (RWEQ) model to quantitatively estimate the sand fixation service based on wind erosion and wind-induced sediment transport between the soil surface and a height of 2 m for specified periods based on a single event [74]. This model was characterized by both empirical and process modeling, which comprehensively considers climate, surface vegetation, surface roughness, soil erodibility, soil crust, and other factors, and thus has been extensively tested under broad field conditions [53,86,87]. The RWEQ involved basic equations, as follows [86,88]:

$$SF = SL_p - SL_a \tag{12}$$

$$Q_{maxp} = 109.8[WF \times EF \times SCF \times K'] \tag{13}$$

$$S_p = 150.71(WF \times EF \times SCF \times K')^{-0.3711} \tag{14}$$

$$SL_p = \frac{2z}{S_p} Q_{maxp} \cdot e^{-(z/S_{\mathrm{p}})} \tag{15}$$

where *SF* represents sand fixation (kg·m$^{-2}$), which is the difference between the amount of potential soil erosion without vegetation cover (*SL*$_P$, kg·m$^{-2}$) and the actual soil erosion under the current land cover and management conditions (*SL*$_a$, kg·m$^{-2}$) per unit area by wind [88]. $Q_{max}$ is the maximum transport capacity (kg·m$^{-1}$); *WF* is the weather factor (kg·m$^{-1}$); *EF* is the soil erodibility factor (%); *SCF* is the soil crust factor (dimensionless); *K'* is the surface roughness (dimensionless) caused by the topography on the wind erosion, which is calculated using terrain data [89]; $S_0$ is the field length scale (m); *Z* is the distance from the upwind edge of the field (m); and *SL* (kg·m$^{-2}$) is the soil loss caused by wind erosion.

$$Q_{maxa} = 109.8[WF \times EF \times SCF \times K' \times VCF] \tag{16}$$

$$S_a = 150.71(WF \times EF \times SCF \times K' \times VCF)^{-0.3711} \tag{17}$$

$$SL_a = \frac{2z}{S_a} Q_{maxa} \cdot e^{-(z/S_a)} \tag{18}$$

where *VCF* is the vegetation coverage factor, which has a significant influence on sand or soil erosion by wind, which are generally estimated by the normalized difference vegetation index [89]; *S* is the critical field length, at which 63% of the maximum transport capacity occurs.

The weather factor (WF) represents the impact of the climate conditions on wind erosion, which is estimated using the following equation [87]:

$$WF = \frac{\sum_{i=1}^{N} U_2(U_2 - U_t)^2 \times N_d}{N} \times \frac{\rho}{g} \times (SW) \times SD \tag{19}$$

$$SW = \frac{ET_p - (R+I)\frac{R_d}{N_d}}{ET_p} \tag{20}$$

$$ET_p = 0.0162(\frac{SR}{58.5})(DT + 17.8) \tag{21}$$

$$SD = 1 - P(\text{snow cover} > 25.4 \text{ mm}) \tag{22}$$

where *WF* is the weather factor (kg·m$^{-1}$), $U_2$ is the wind speed at 2 m (m·s$^{-1}$), $U_t$ is the threshold wind speed at 2 m (assumed (m·s$^{-1}$)), *N* is the number of wind speed observations (normally 500), $N_d$ is the number of days in the time period, $\rho$ is the air density (kg·m$^{-3}$), *g* is the acceleration due to gravity (m·s$^{-2}$), *SW* is the soil wetness (dimensionless), and *SD* is the snow cover factor (dimensionless). *ET$_P$* is the potential relative evapotranspiration (mm), (*R* + *I*) is rainfall and irrigation (mm), $R_d$ is the number of rainfall and/or irrigation days, SR is solar radiation (cal·cm$^{-2}$), *DT* is the average temperature (°C), and *P* is probability of snow depth more than 25.4 mm.

The soil erodible fraction (EF) and the soil crust factor (SCF) depend on the soil texture, and are estimated by the following equations [74]:

$$EF = (29.09 + 0.31Sa + 0.17Si + 0.33Sa/Cl - 2.59OM - 0.95CaCO_3)/100 \tag{23}$$

$$SCF = 1/(1 + 0.0066(Cl)^2 + 0.021(OM)^2) \tag{24}$$

where *Sa* is the sand content (%), *Si* is the silt content (%), *Sa/Cl* is the sand-to-clay ratio, OM is organic matter (%), and $CaCO_3$ is the calcium carbonate content (%).

### 2.3.7. Statistical Analyses

Pearson correlation coefficient analysis was used to examine the relationships between the ecosystem services; positive correlation implies a synergistic relationship between two ecosystem services, and negative correlation implies a certain trade-off between the paired ecosystem services [25]. The *p* value was used to detect significant differences between the ecosystem services; * and ** mean significance at the $p < 0.05$ and $p < 0.01$ levels, respectively.

Based on the daily observations from 1980 to 2018, the annual change rates of the temperature, precipitation and wind speed were calculated using linear regression. Based on the principles of reliability, continuity and accessibility, the vegetation cover was calculated using the NDVI dataset from 2000 to 2020 from the Google Earth Engine (https://ladsweb.modaps.eosdis.nasa.gov/search/, accessed on 10 August 2021). The Sen's slope was used to estimate the NDVI change per unit of time [90]. The statistically significant changes in the annual temperature, precipitation, wind speed and vegetation cover from 1980 to 2018 were detected by the nonparametric Mann-Kendall method [91].

Redundancy analysis (RDA) was applied for the identification of the main environmental factors influencing the ecosystem services [25]. RDA is a canonical analysis method combining regression analysis and principal component analysis, which is appropriate to regress several explanatory variables (i.e., the potential drivers) against multiple response variables (i.e., the six ecosystem services) [33,92].

Spatial autocorrelation generally includes global and local spatial autocorrelation, which measure the degree of aggregation or dispersion between the attributes of spatial elements [41,93]. The global spatial autocorrelation is an overall characteristic of the spatial pattern, and does not reflect the location of the clusters. In order to identify local clusters and spatial outliers, Anselin [93] developed the Local Moran's I, which is also known as the local indicators of spatial association (LISA), which is a kind of local spatial autocorrelation. LISA mainly includes univariate and bivariate Local Moran's I, especially bivariate Local Moran's I, which has become an effective method to study the spatial distribution of different geographical elements [28,94]. More specifically, for bivariate Local Moran's I analysis including four different types of spatial clusters, high–high and low–low clusters indicate that the associations are positive, and are described as synergistic relationships; high–low and low–high clusters indicate that the associations are negative, and are described as trade-off relationships. The above four clusters are significant at $p = 0.05$; non-significant correlation indicates no obvious trade-off and synergy relationship [94]. In order to understand the spatial trade-off/synergy relationship of different ecosystem services in the Hexi region, this research used 1 km × 1 km fishnets as the basic unit to assign the statistical results of ecosystem services to the vector layer, and then imported GeoDA software, which created bivariate LISA statistics for the ecosystem services. The bivariate spatial autocorrelation method was employed to reveal the spatial heterogeneity of the trade-offs and synergistic relationships among the ecosystem services, which also provide a scientific basis for the understanding of the spatial variability of ecosystem services in the Hexi Region. We resampled all of the raster data to a 100-m resolution in order to allow for uniform simulations. All of the statistical analyses were performed using R software.

### 3. Results

*3.1. The Spatiotemporal Changes of the Ecosystem Services*

For carbon storage, water retention, soil retention, food production and habitat quality—unlike sand fixation—there were similar spatial distribution patterns in the Hexi Region from 1980 to 2018, which were relatively stable, and their local variations were more obvious (Figures 4–9). All of the ecosystem services significantly increased over the past four decades, with the exception of sand fixation and habitat quality for biodiversity (Figure 10). In the Qilian Mountains, food production, water retention, soil retention and carbon storage services increased by $2075.40 \times 10^4$ t, $84.56 \times 10^8$ m$^3$, $279.43 \times 10^8$ t, and $0.22 \times 10^8$ t, respectively, whereas sand fixation decreased by $190 \times 10^4$ t (Figure 10). Furthermore, food production was characterized by the largest increase rate ($10.27\% \cdot yr^{-1}$) in the Qilian Mountains, followed by water retention ($4.98\% \cdot yr^{-1}$), soil retention ($3.60\% \cdot yr^{-1}$), carbon storage ($0.03\% \cdot yr^{-1}$), and habitat quality ($0.04\% \cdot yr^{-1}$), whereas sand fixation decreased at a decreasing rate of $1.30\% \cdot yr^{-1}$. In the Hexi Corridor, food production, water retention, soil retention and carbon storage increased by $864.53 \times 10^4$ t, $2.61 \times 10^8$ m$^3$, $8.41 \times 10^8$ t, $0.22 \times 10^8$ t, respectively, whereas sand fixation decreased by $178.96 \times 10^4$ t and habitat quality decreased slightly. Over the past four decades, food production had the largest increase ($9.93\% \cdot yr^{-1}$), and yet the largest reduction in sand fixation was 0.79% per year in the Hexi Corridor.

Specifically, the spatial distribution of carbon storage, habitat quality, food production, water retention and soil retention showed an overall decreasing trend from the southeast to northwest in the Hexi Region (Figures 4–8); the high-value regions and increasing regions of the first three ecosystem services were similarly located; both aggregated in the eastern part of the Qilian Mountains and the oases of the Hexi Corridor. Moreover, the high-value regions for water retention and soil retention were located in the Qilian Mountains; the low-value regions were located in the Hexi Corridor. For nearly 40 years, it has been noteworthy that the spatial distribution of the sand fixation was relatively stable (Figure 9), and approximately 90% or more of the total area remained unchanged; the high value and increasing areas of sand fixation were more consistent, and were mainly distributed in the west of the Hexi Region, specifically in the southwest of the Qilian Mountains and the northwest of the Hexi Corridor; not all of the regions had a positive trend, and the decreasing areas of sand fixation were mainly concentrated in the oasis of the Hexi Corridor and the upstream of the Qinghai Lake basin. Overall, our results suggested that, except for sand fixation, all of the ecosystem services in the Qilian Mountains were greater than those in the Hexi Corridor, and water retention and soil retention services were mainly concentrated in the eastern Qilian Mountains with extensive forest, shrub and medium to high coverage grassland. Increased food production was mainly clustered in the eastern Qilian Mountains and in the oases of the Hexi Corridor. The increasing region of sand fixation was mainly distributed in the west of the Hexi Region, where deserts are widespread and wind–sand weather occurs frequently.

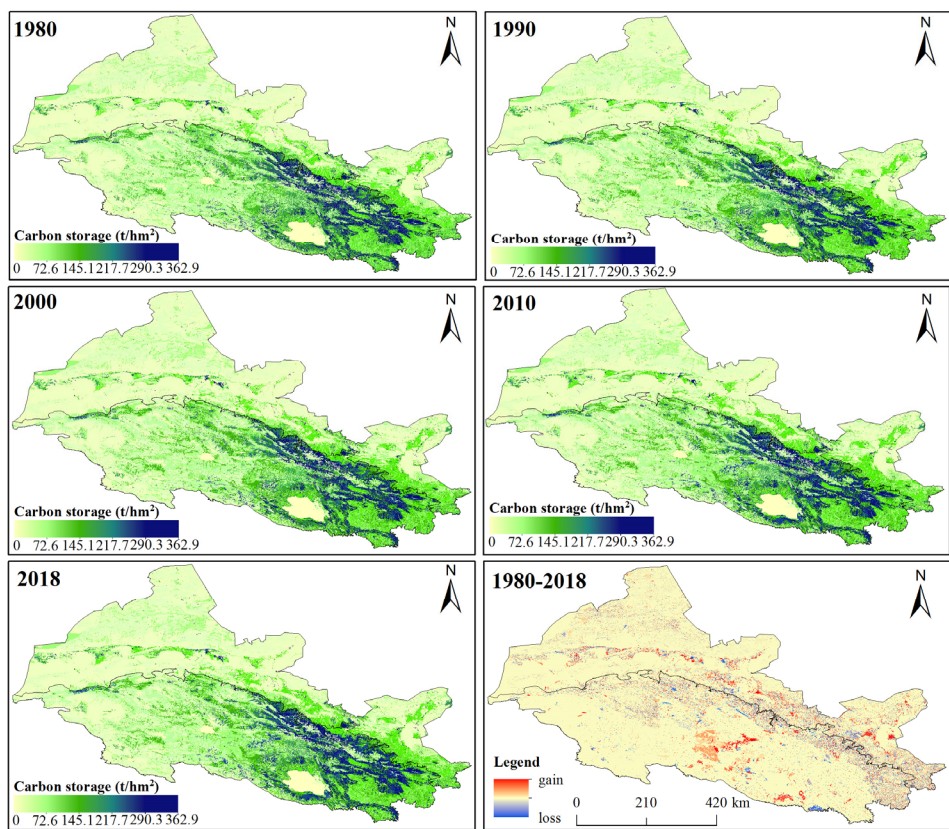

**Figure 4.** Spatial distribution of carbon storage from 1980 to 2018 in the Hexi Region.

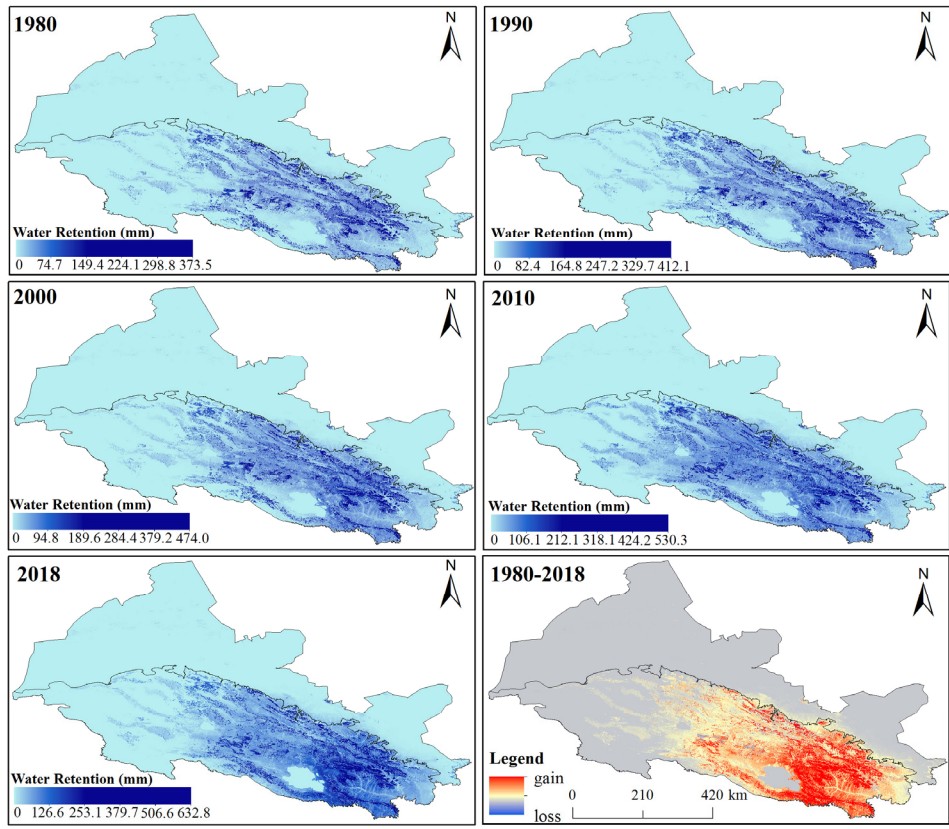

**Figure 5.** Spatial distribution of water retention from 1980 to 2018 in the Hexi Region.

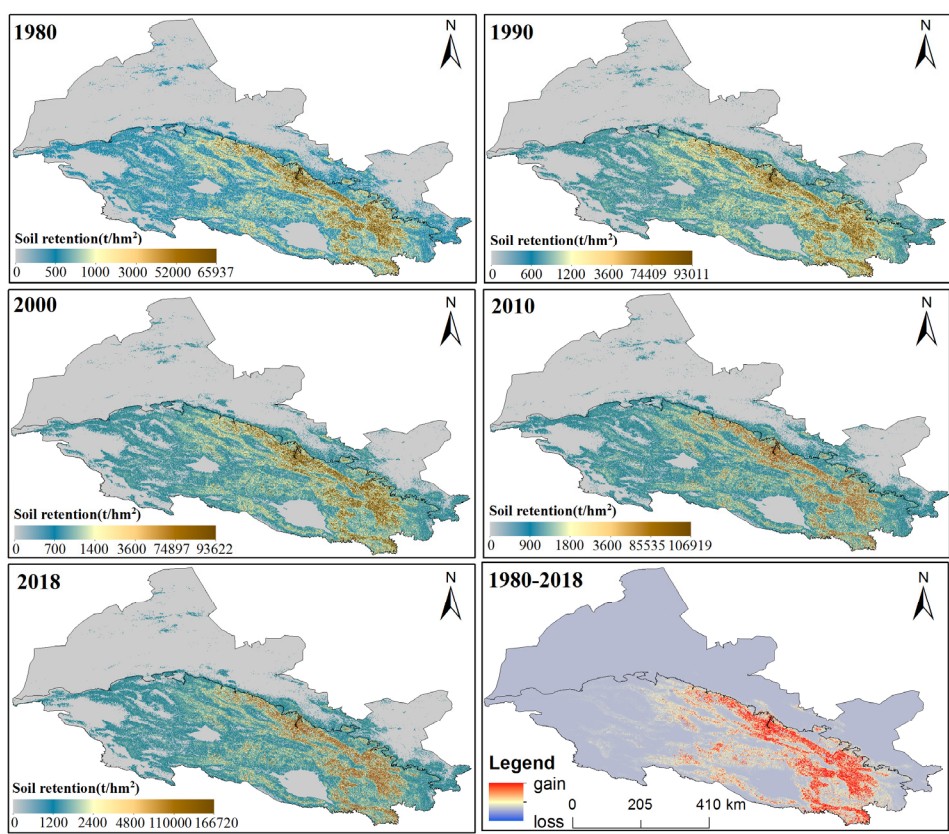

**Figure 6.** Spatial distribution of soil retention from 1980 to 2018 in the Hexi Region.

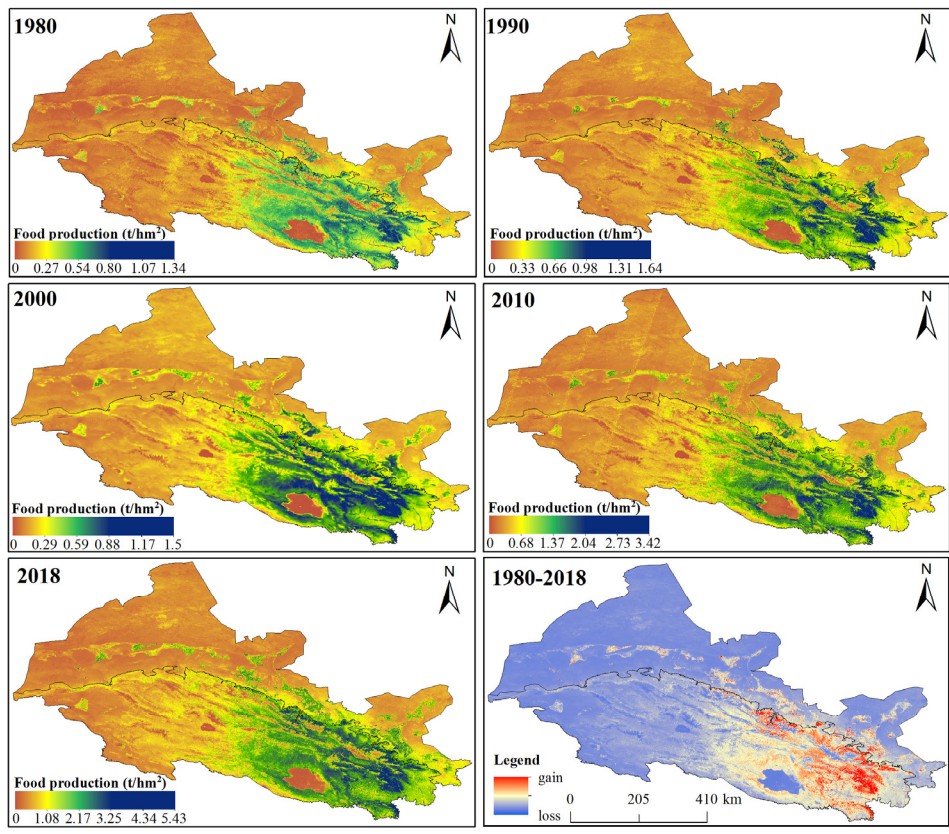

**Figure 7.** Spatial distribution of food production from 1980 to 2018 in the Hexi Region.

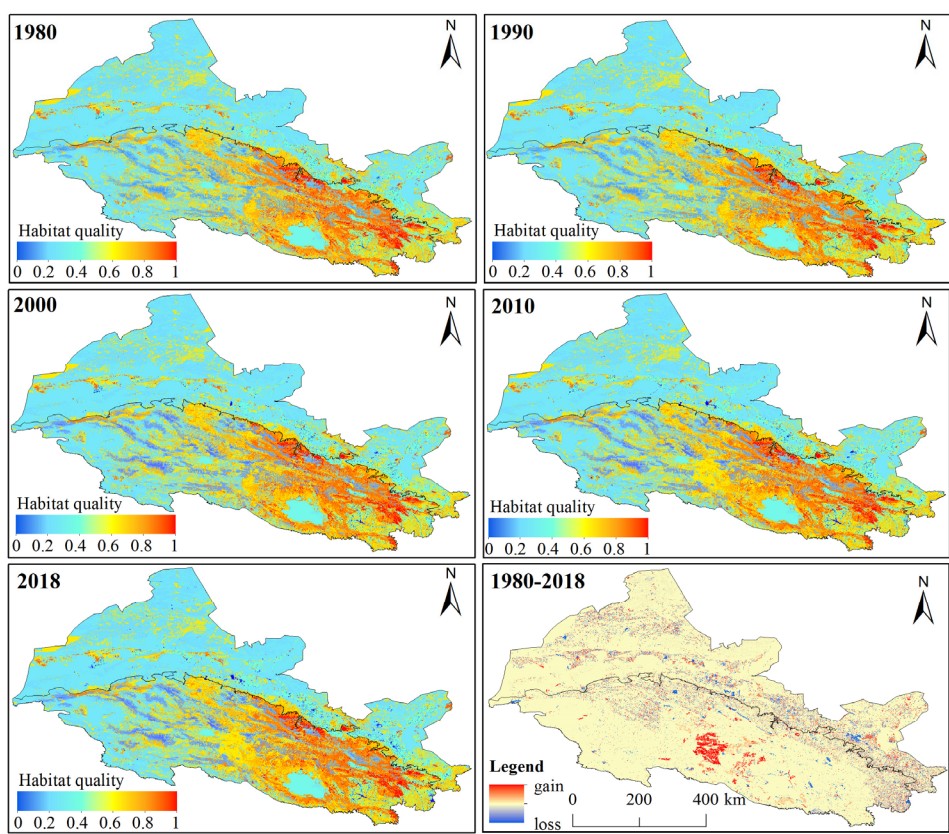

**Figure 8.** Spatial distribution of habitat quality from 1980 to 2018 in the Hexi Region.

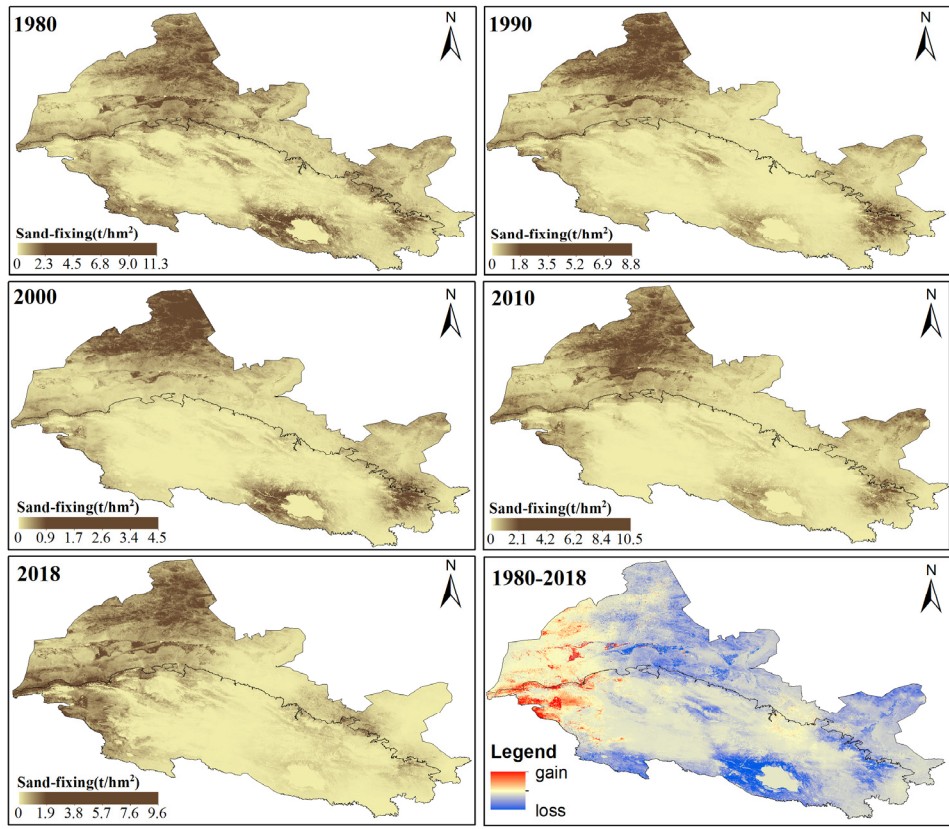

**Figure 9.** Spatial distribution of sand fixation from 1980 to 2018 in the Hexi Region.

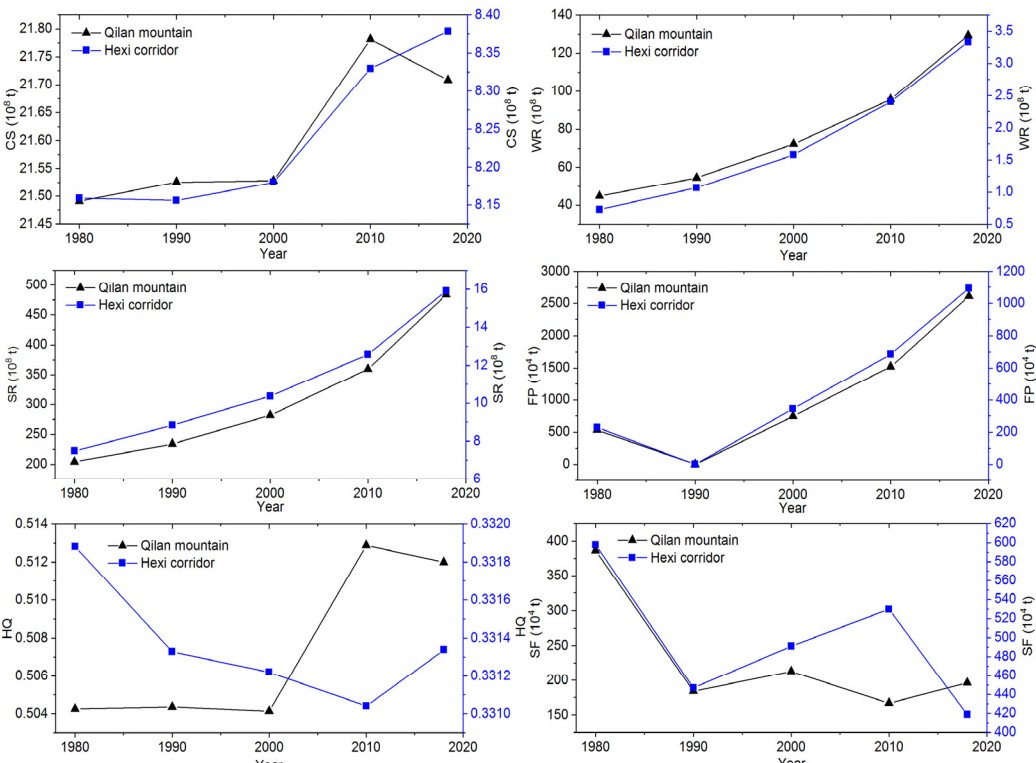

**Figure 10.** Temporal changes of the ecosystem services in the Hexi region from 1980 to 2018. CS: Carbon Storage; WR: Water Retention; SR: Soil Retention; FP: Food Production; HQ: Habitat Quality; SF: Sand Fixation.

### 3.2. Trade-Offs and Synergies among Ecosystem Services

In order to reveal the steady trade-offs and synergies between ecosystem services, the correlation relationships were detected based on the multi-year averages of six ecosystem services. The results showed that, except for sand fixation, other ecosystem services in the Hexi Corridor had significant synergistic relationships with each other (Figure 11a), and carbon storage has a strong synergistic relationship with food production and habitat quality; sand fixation was negatively correlated with food production, water retention and soil retention. In the Qilian Mountains, the relationships between the ecosystem services were dominated by strong synergistic relationships and weak trade-offs (Figure 11b). Among them, there were strong synergistic relationships with carbon storage, food production and habitat quality, similar to that of the Hexi Corridor. Moreover, water retention also had a strong synergistic relationship with food production and soil retention. There was a somewhat-weak trade-off between sand fixation, water retention and soil retention. In short, synergistic relationships were the dominant relationships between the ecosystem services in Hexi Region. The synergistic relationships in the Qilian Mountains were stronger than those in the Hexi Corridor, while the trade-off relationships were the opposite.

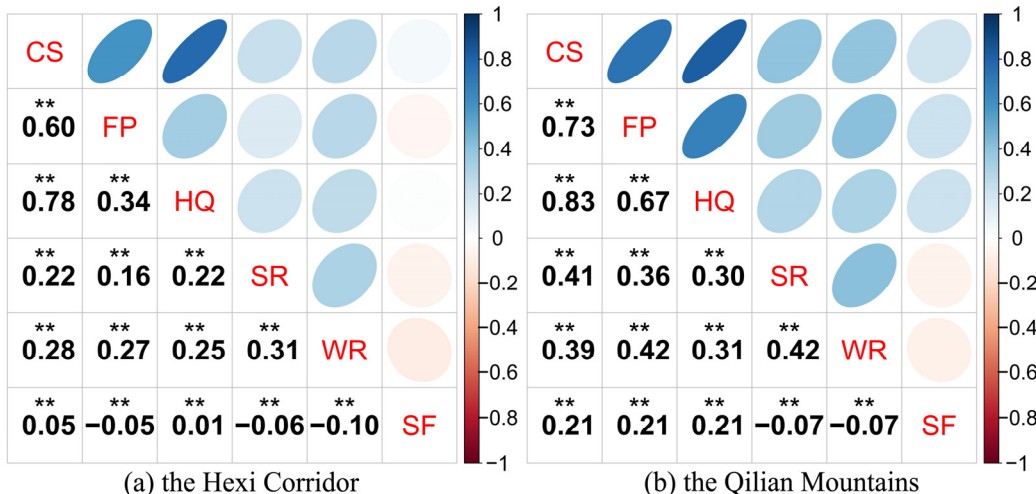

**Figure 11.** Pearson correlations between pairs of ecosystem services in the Hexi Region, China. CS: Carbon Storage; FP: Food Production; HQ: Habitat Quality; SR: Soil Retention; WR: Water Retention; SF: Sand Fixation. The blue and red colors indicate positive and negative correlations, respectively. Meanwhile, the Pearson correlation coefficient and its significance level are shown under the diagonal. Note: ** indicate that the correlations are significant at the 0.01 levels.

As shown in Figure 12, spatially, the relationships of the six ecosystem services were mainly characterized by synergistic relationships between high–high clusters and low–low clusters in the Hexi Region, with significant spatial heterogeneity. Furthermore, the synergistic relationship of the high–high cluster in the ecosystem services (i.e., water retention with habitat quality, food production and carbon storage) was largely concentrated in southeast of the Qilian Mountains, in which it provides rich water resources, high vegetation coverage and food supply. For instance, higher carbon storage was associated with higher habitat quality and food production in the Qilian Mountains, which form approximately 30% of the whole study area. Compared with the mountains, the Hexi Corridor, at low elevations, has been through drought and water shortage with sparse vegetation, and exhibited the low–low agglomeration of a synergistic relationship among ecosystem services. Beyond that, there was a significant synergy between carbon storage and food production in the oases in the Hexi Corridor, showing mainly high–high concentrations. Additionally, water retention, soil retention, habitat quality, food production and carbon storage have markedly spatial trade-offs with sand fixation in the Hexi region (Figure 12). In the Qilian Mountains, the trade-offs were primarily distributed in the central and eastern areas of the mountains, with a high proportion of high–low aggregation for water retention and sand fixation, followed by habitat quality and sand fixation. In other words, abundant water sources and higher vegetation coverage in this region have brought better habitat quality, resulting in lower sand fixation services in these areas. The trade-off relationships in the Hexi Corridor accounted for up to 16.73% of the whole study area, and were mainly distributed in the northwest of this region, which is dominated by low–high aggregation. This is to say that the lack of water resources and sparse vegetation have led to poorer habitat quality and other lower other ecosystem services, in combination with strong wind erosion, which may contribute to the higher amount of sand fixation. Given that the agriculture irrigation of oases in the Hexi Corridor relies on upstream water resources from the Qilian Mountains, food production has obvious trade-offs with habitat quality, water retention and soil retention.

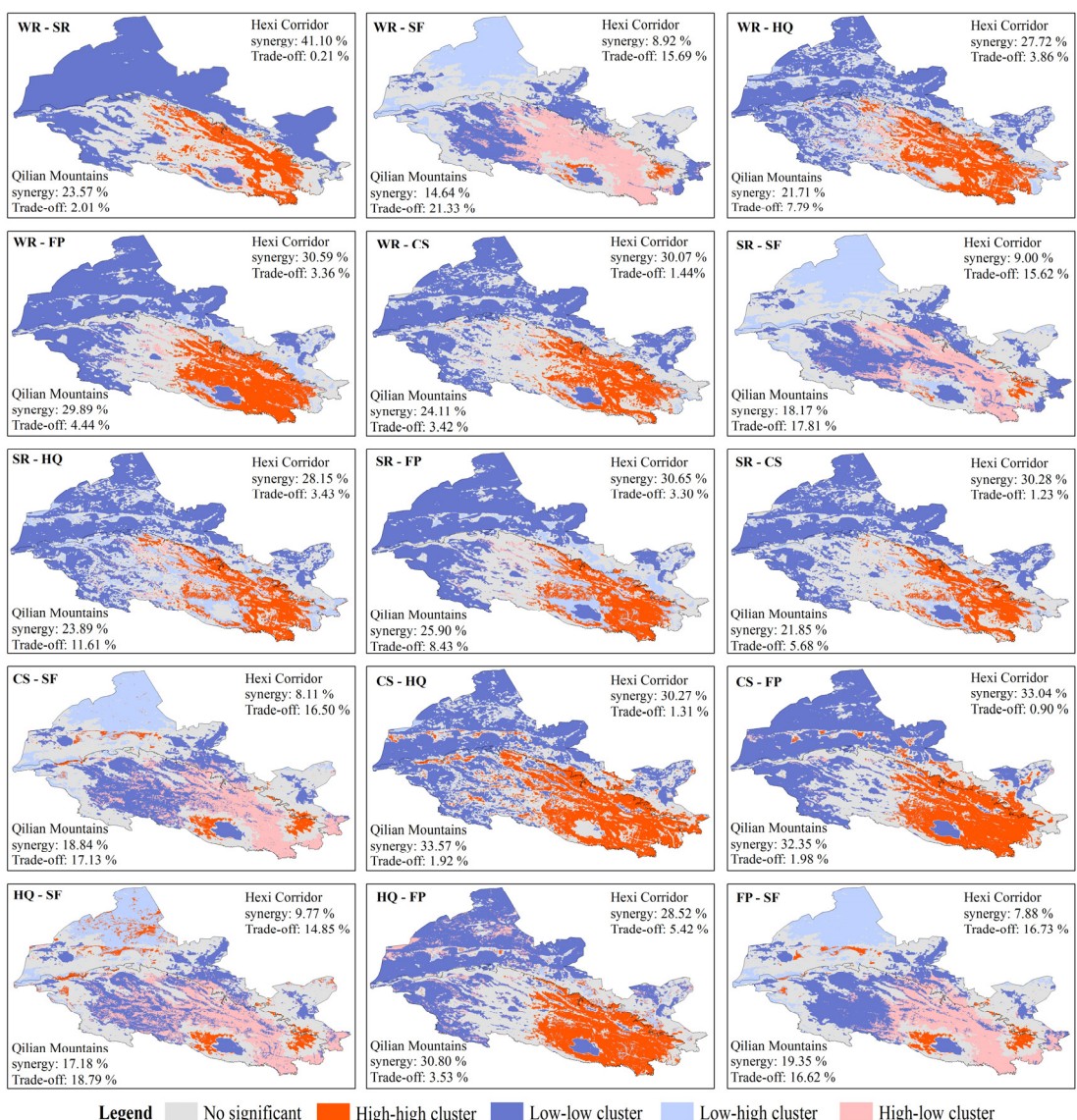

**Figure 12.** LISA cluster map of the ecosystem services in the Hexi region from 1980 to 2018. WR: Water Retention; SR: Soil Retention; SF: Sand Fixation; CS: Carbon Storage; FP: Food Production; HQ: Habitat Quality. High–high clusters and low–low clusters indicate synergistic relationships, and high–low clusters and low–high clusters indicate trade-off relationships. The numbers indicate trade-offs and synergies as a percentage of the overall study area.

### 3.3. Drivers of Ecosystem Services

Figure 13 displays the biplots of the RDAs, which were performed in order to identify the driving factors of the changes in the ecosystem services. In the Hexi Corridor, the results demonstrated that the explanatory variables of RDA accounted for 43.27% of the variance in the plains, and RDA1 and RDA2 explained 35.08% of the variance (Figure 13a); the mean annual wind speed, potential evapotranspiration, mean annual temperature and precipitation were the key factors affecting the sand fixation in the Hexi Corridor, especially the wind speed factor; slope, mean annual precipitation, rainfall erosivity index and fractional vegetation cover were crucial factors that impacted soil retention and water retention. In addition to the mean annual precipitation and fractional vegetation cover, population and cropland expansion played a key role in carbon storage, habitat quality and food production services. Furthermore, this study also showed that the population, and

farmland effective irrigated area from 1999 to 2018 in the Hexi Corridor increased significantly, and the total water resources showed an increasing trend, while this trend was not significant at the 0.05 level (Figure 14).

In the Qilian Mountains, the explanatory variables of RDA accounted for 54.51% of the variance, and the first two canonical axes explained 48.51% of the variance (Figure 13b); the fractional vegetation cover, mean annual precipitation, rainfall erosivity index, and potential evapotranspiration were key factors affecting carbon storage, food production, and habitat quality, with potential evapotranspiration being significantly and negatively correlated with these three ecosystem services. The pivotal factors for the sand fixation in these mountains were the wind speed, temperature, altitude, and potential evapotranspiration. Soil retention and water retention services were mainly determined by the slope, mean annual precipitation, rainfall erosivity index and vegetation cover. Therefore, the role of natural factors for ecosystem services was dominant in the Qilian Mountains. In recent decades, the vegetation cover, precipitation and temperature in the Hexi Region have increased significantly (Figure 15), and this has led to improvements in the water retention, carbon storage and habitat quality, which further enhance soil retention. At the same time, with the wind speed having decreased significantly, the amount of sand fixation also decreased in this region.

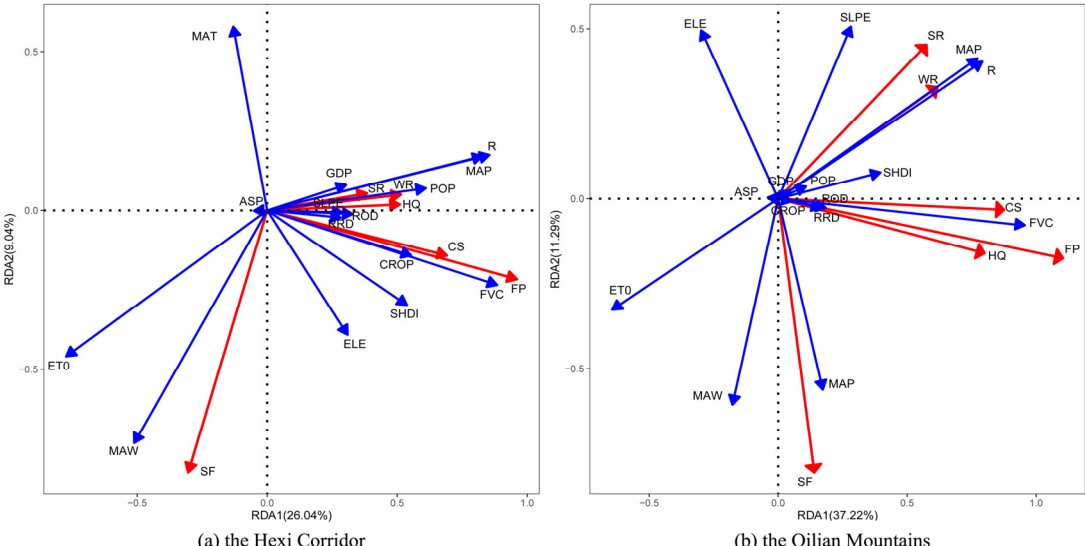

(a) the Hexi Corridor        (b) the Qilian Mountains

**Figure 13.** RDA biplots depicting the relationships between the ecosystem services and their drivers. Note: Red arrows with red text next to them represent the six ecosystem services; blue arrows with blue text next to them represent the fifteen factors, and the length of blues arrows represents the contribution of the driving factors to the ecosystem services. The cosine of the angle between the ecosystem services and the driving factor arrows reflects the correlation. The six ecosystem services—WR: Water Retention; SR: Soil Retention; SF: Sand Fixation; CS: Carbon Storage; FP: Food Production; HQ: Habitat Quality. The fifteen drivers are the elevation (ELE), aspect (ASP), slope (SLPE), potential evapotranspiration (ET0), mean annual temperature (MAT), mean annual precipitation (MAP), mean annual wind speed (MAW), fractional vegetation cover (FVC), rainfall erosivity index (R), Shannon's diversity index (SHDI) and river network density (RRD), and road density (ROD), population size (POP), gross domestic product (GDP), and cropland area (CROP). The RDA analysis passed the permutation test and significance test ($p < 0.001$).

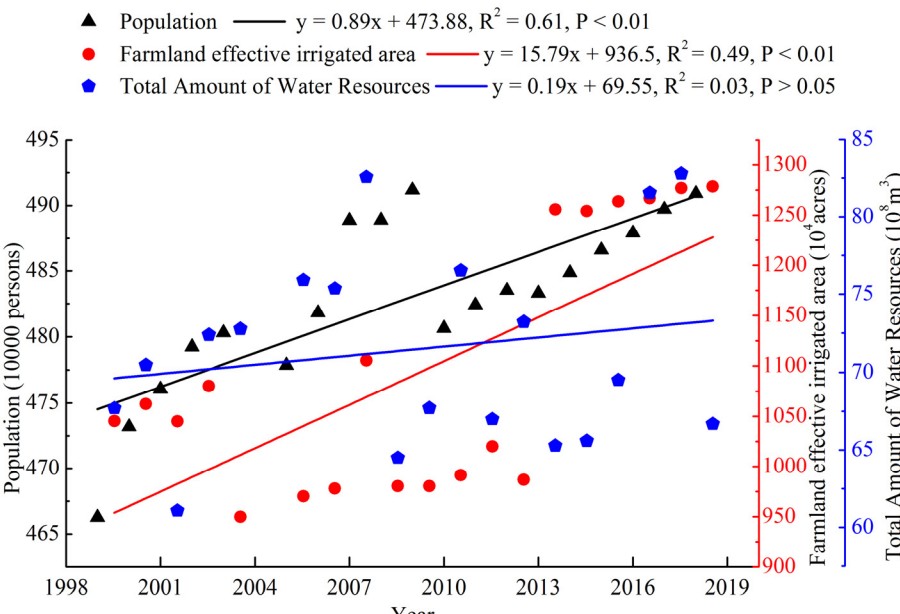

**Figure 14.** Changes in the population, farmland effective irrigated area, and total amount of water resources from 1999 to 2018 in the Hexi Corridor.

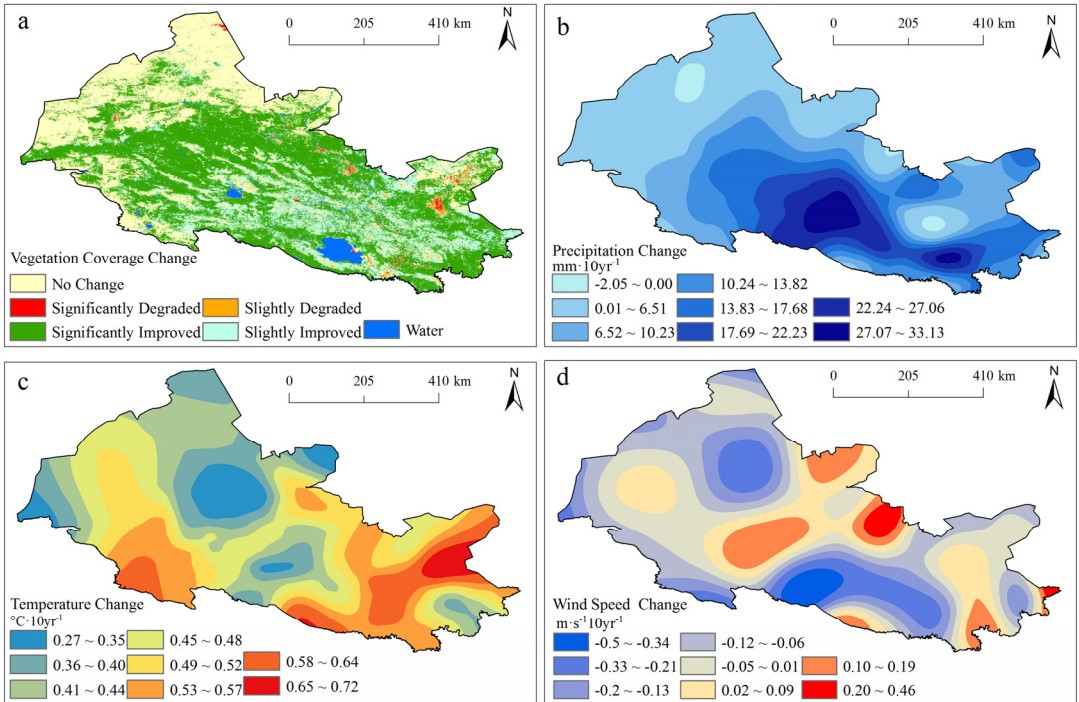

**Figure 15.** Change trend of the annual vegetation coverage (**a**), precipitation (**b**), temperature (**c**) and wind speed (**d**) in the Hexi Region of northwestern China from 1980 to 2018. Note: Based on the principles of reliability, continuity and accessibility, the time series selected for the vegetation cover data is from 2000 to 2020.

## 4. Discussion

### 4.1. Changes of the Ecosystem Services in the Different Regions

The spatial distribution of the carbon storage, habitat quality, food production, water retention and soil retention were characterized by strong spatial heterogeneity, and an overall decreasing trend from southeast to northwest over the Hexi Region, which was consistent with the previous studies. Xu et al. [26] emphasized that the relatively low carbon storage, soil retention, water yield and habitat quality were mainly located in the areas with the lower vegetation coverage due to insufficient rainfall and intense evapotranspiration, such as northwest China, Mongolia, Central Asia and Western Asia compared to the East Asia and Southeast Asia. Moreover, our results demonstrated that all of the ecosystem services from 1980 to 2018 in the Qilian Mountains with the higher vegetation coverage and more precipitation were greater than those in the Hexi Corridor, apart from sand fixation services. Kang et al. [88] found that mountains had lower sand fixation than deserts and oases, and the highest value of the sand fixation was concentrated on the Alxa Plateau–Hexi Corridor of the northwestern arid area in China during 1990–2015. Furthermore, the five ecosystem services have been enhanced over the past 40 years in the whole region, with the exception of sand fixation in this study. Previous studies have suggested that ecosystem services (e.g., food production, carbon storage, water retention and soil retention) have improved over the past few decades in China, apart from habitat provision [25,32,39]. For example, the annual average value of the carbon storage, soil retention and water retention per unit area increased by 0.38%, 84% and 147.5%, respectively, from 2000 to 2010 in an arid inland river basin in northwest of China [31]. Hua et al. [57] also pointed out that the carbon sequestration and habitat quality experienced significant growth on the Tibetan Plateau of China. The sand fixation generally experienced a decline in the Hexi Region over the past few decades, and similar results were also obtained in this study. The sand fixation in the arid region of northwest China showed a remarkable downward trend, with a reduction of 3.67 t·hm$^{-2}$ in the last 25 years [88].

### 4.2. The Trade-off and Synergy Relationship of Ecosystem Services

In order to better understand the spatial heterogeneity of the relationships among the ecosystem services, our research explored the relationships between the pair-wise ecosystem services of the Qilian Mountains and Hexi Corridor in the Hexi Region. Based on quantitative and spatial correlation analysis, we found that the relationships among the ecosystem services were mainly characterized by strong synergies and weak trade-offs with significant spatial heterogeneity in the Hexi Region, and the synergistic and trade-off relationships in the Qilian Mountains were stronger than those in the Hexi Corridor (Figures 11 and 12). These were consistent with the results observed [95]. Specifically, there were significant and synergistic relationships with water retention, soil retention, habitat quality, food production and carbon storage which were largely concentrated in the southeast of the Qilian Mountains and the northwest of the Hexi Corridor. The enhanced water retention service contributed to the growth of vegetation, improved the surrounding ecological environment, and promoted the further improvement of the soil and water conservation capacity. These were similar to the findings of previous studies in other regions. For example, Gou et al. [32] found that synergies occurred between carbon storage and habitat quality, and carbon storage and soil retention in the Three Gorges Reservoir Area of the upper and middle Yangtze River. There were stably synergetic relationships between soil retention and grain production, water yield and soil retention, and water yield and grain production in Qinling-Daba Mountain of the midwestern area of China [83]. Between food provision and carbon storage, and carbon storage and water retention, there were marked synergies in the arid inland basin of northwest China, with limited water resources [31]. Of course, there were some differences between our results and other studies in arid regions due to the different scales and the geographical environ-

ment. For example, Wang et al. [96] pointed out that the relationship between carbon storage and water retention was a trade-off due to afforestation causing the wastage of water resources [96].

Additionally, the complexity of ecosystems and drivers has led to synergies and trade-offs among various ecosystem services. Water retention, soil retention, habitat quality, food production and carbon storage have marked spatial trade-offs with sand fixation in the central and eastern Qilian Mountains and the northwestern Hexi Corridor. Previous studies have shown that sand fixation has weak trade-offs with food production and carbon sequestration in Ningxia of the Yellow River Basin, China [97]. Over the past 25 years, soil retention and sand fixation have had a weak relationship in the northwestern arid Area of China [88]. Moreover, due to the arid climate and higher land-use intensity in the oases of the Hexi Corridor, agricultural activities have relied on water sources from the Qilian Mountains, so there were remarkable trade-offs between habitat quality and food production, food production and soil retention, and food production and water retention in this plain. Existing research has also highlighted that the trade-off relationship of provisioning services with biodiversity and ecosystem functions strengthened at higher land-use intensity levels [13].

### 4.3. Driving Factors of Ecosystem Services

Our study has indicated that the changes and relationships of ecosystem services depend on both natural environmental and socioeconomic factors in the Hexi Region (Figure 11). Climate change has driven synergies and tradeoffs among the ecosystem services in the entire study region, which was in accordance with some results [34,36,57]. For example, stronger carbon storage capacity was usually associated with higher vegetation cover. In recent years, many researchers have indicated that climate warming and humidifying were the primary reasons for the improved vegetation coverage in the Hexi Region [98–100]. In particular, due to ecological restoration and the warming and humidification of the climate, vegetation coverage has improved significantly in most regions except for some urban, rural, industrial, mining and residential land (Figure 15), which was in accordance with other studies [101,102]. In addition, relevant studies have revealed that the surface wind speed significantly decreased in China during the past 50 years due to climate warming exacerbated by the weakening large-scale thermal differences [100,103]. As the temperature and precipitation have increased significantly, and wind speed has markedly decreased in these regions over the recent decades (Figure 15), and these have led to enhancement in runoff and thus water resources and vegetation coverage, which further enable the improvement of water retention, habitat quality, carbon storage, food production and soil retention, and sand fixation reduction.

Generally, similar driving mechanisms and significant synergies existed among food production, carbon storage and habitat quality in the Qilian Mountains. Moreover, due to the fragile alpine ecosystems coupled with slow human disturbance in the Qilian Mountains, natural environmental factors have played major roles in ecosystem services. Soil erosion generally happened in mountainous regions featuring steep terrain and strong variation in rainfall and runoff [104]. The slope and rainfall erosivity index had the dominant impact on soil retention, which kept increasing with precipitation and vegetation coverage [105]. Water retention services were mainly determined by the rainfall erosivity index, which increased continuously with the rainfall and vegetation coverage. Similar drivers allowed for an obvious synergistic relationship between soil retention and water retention in most parts of the Qilian Mountains, which was similar to other regions [106,107]. The key factors for sand fixation were the wind speed, temperature and altitude in these mountains. A similar study highlighted that slope had the dominant impact on sand fixation, and kept increasing with elevation and wind speed in northwestern China [97]. Water was the radical limiting factor in the Hexi Corridor for vegetation growth and social development, which has relied primarily upon meltwater from glaciers and snow in the Qilian Mountains. Besides this, wind speed and potential evapotranspiration were

the crucial factors affecting the sand fixation in the Hexi Corridor, especially the wind speed factor, which was similar to that in the Qilian Mountains.

Immense socioeconomic development and rapid urbanization have influenced multiple ecosystem services synchronously, and have further affected the relationships among ecosystem services [107,108]. The driving factors of the synergistic and trade-off relationships in the Hexi Corridor were different from those in the mountains. Besides precipitation and vegetation cover, the population growth and cropland expansion also drove a synergy between carbon storage and food production, and trade-offs between food production and other ecosystem services in the Hexi Corridor, as they increased cropland area and water consumption, which was consistent with a previous study [109]. The initial carbon storage in the arid-desert regions was very low as a result of the limited water, while croplands based on desert reclamation were well irrigated, and carbon storage was positively correlated with food production in the Hexi Corridor due to the greater belowground biomass input than that in the deserts [110]. However, the intensification of the trade-off relationships between food production and habitat quality, and water retention and soil retention in the oases of the Hexi Corridor, which were related to population growth and continuous cropland expansion, exacerbated the conflict between agricultural and ecological water use. The relevant research also demonstrated that the remarkable trade-off between crop production and habitat quality occurred at altitudes of less than 0.5 km in the Belt and Road region due to the increase in population promoted the higher food production and the expansion of cropland, and occupied a mass of natural habitats, which caused the decline of biodiversity [26,105]. Additionally, the construction of artificial lakes has increased water availability for drinking and irrigation in semi-arid landscapes, as well as the expansion of the agricultural area and a decrease in water purification and sediment retention services [111]. Therefore, the appropriate water resource management policies in the arid region have been essential to sustain ecosystem services, maintaining a balance between conflicting demands from agriculture development and ecological protection [112].

*4.4. Management Implications*

The relevant research has emphasized the importance of managing larger regions by analyzing the spatiotemporal characteristics, relationships and drivers of the ecosystem services for the improvement of the ecosystem services and human wellbeing [113]. Ecosystem services in the Hexi region have generally improved over the past few decades, but the locals still faced some challenges in different regions. The existing research revealed that mountains—as refuges for biodiversity—may be likely to be threatened by dramatically ongoing global changes in climate and land use [114]. The challenge in the Qilian Mountains is to sustain the synergistic relationship among ecosystem services. As grassland was the main land type in the Qilian Mountains (Figure 2), grazing prohibition has been an inevitable important means for the locals to remediate degraded grassland and protect the ecological environment. The problem of balancing long-term ecological conservation with herders' livelihoods in these mountains has also existed since the establishment of the Qilian Mountains National Nature Reserve in the 1980s. Moreover, all the time, grazing exclusion with fences has been an effective way to restore degraded grasslands in alpine mountains and elsewhere [37]. However, some studies have emphasized that the longer-term fencing over 8 years has hindered wildlife movement, increased grazing pressure in unfenced areas, and expended substantial financial costs to the governments on the Tibetan Plateau [115,116]. The locals should optimize grazing exclusion practices which avoid fencing in key wildlife habitat regions, especially the protected large mammal species [115]. In addition, there were the trade-offs between food production and sand fixation, water retention, soil retention, and habitat quality in the Hexi Corridor with limited water, which may be further strengthened in longer time scales due to climate warming causing the continuous glacier shrinkage and glacial meltwater to in-

crease and then decrease. Water management ought to focus more on the rational utilization of water resources to ensure ecological water demand for the locals. For example, highly efficient water-saving irrigation and crops enable us to save extra water resources for further ecological restoration in desert regions.

*4.5. Limitations of the Study*

Although six ecosystem services were calculated in the Hexi Region, there are still many opportunities and challenges in the data acquisition. Over the past few decades, remote sensing with the option of fast, frequent, and continuous observations has shown increased utility for environmental monitoring and biodiversity conservation at spatial scales [55,56]. In particular, high-spatial-resolution datasets based on remote sensing inversion, as proxy indicators, have provided new opportunities for monitoring ecosystem services at a finer spatial scale than they previously did. Remote sensing data such as soil moisture (Soil Moisture Active Passive or Sentinel-1A), terrestrial evapotranspiration (MOD16A2 products), precipitation (Global Precipitation Measurement), soil properties (SoilGrids based on the state-of-the-art machine learning models and remote sensing-based environmental covariates) and species distribution (Quickbird), should be further employed in the evaluation of diverse ecosystems services in the Hexi Region [56,117]. In addition, there are uncertainties in the assessment of ecosystem services, as not all of the impact factors were considered in the model due to limited data accessibility [82,85], and the parameters required in different models were generally derived from literature results or based on empirical methods or similar regions [32,75]. Furthermore, the more complex models were relatively sensitive to different data sources and resolutions. Thus, on one hand, further research is needed to evaluate the role of these remote sensing-based products in affecting the evaluation precision of ecosystem services; on the other hand, it is also necessary to further consider more ecosystem services and analyze the trade-offs and synergies between ecosystem services and regional responses to global change, based on datasets with higher spatiotemporal resolution and more site-level observations in semi-arid regions of China, and to utilize more long-term and high-resolution data to obtain more reliable results.

**5. Conclusions**

In this study, the spatial distribution of carbon storage, habitat quality, food production, water retention and soil retention showed an overall decreasing trend from the southeast to the northwest in the Hexi Region of China. In particular, all of the ecosystem services in the Qilian Mountains were greater than those in the Hexi Corridor, except for sand fixation. From 1980 to 2018, the majority of the ecosystem services improved in the Hexi Region, but the reduction of sand fixation in the Qilian Mountains was 1.06 times higher than that in the Hexi Corridor. The results also indicated that the relationships among the ecosystem services were mainly characterized by strong synergistic relationships and weak trade-offs with significant spatial heterogeneity in the Hexi Region, and the spatial synergistic and trade-off relationships in the Qilian Mountains were stronger than those in the Hexi Corridor. In addition, food production has significant trade-offs with water retention, soil retention and habitat quality in the oases of the Hexi Corridor. This is mainly because population growth and cropland expansion have exacerbated water scarcity and occupied the natural habitat. Precipitation, temperature and vegetation cover in the Hexi Region have increased significantly over the four past decades, leading to the enhancement of the water retention, carbon storage and habitat quality, which further increased soil retention, especially in the Qilian Mountains. Meanwhile, the amount of sand fixation also decreased with the reduction in wind speed. In the future, the government should further promote the highly efficient utilization of water resources and mitigate trade-offs of the ecosystem services in the Hexi Corridor, and strengthen scientific ecological restoration in the Qilian mountains to further enhance ecosystem services.

Our results highlighted the importance of climate wetting and appropriate cropland expansion in the coordination of trade-offs, in order to ensure the effective management of ecosystem services for arid inland regions.

**Author Contributions:** Conceptualization, Y.L., W.L. and Q.F.; methodology, Y.L., M.Z. and L.Y.; software, Y.L, M.Z. and J.Z.; resources, Q.F., M.Z. and L.Y.; writing—original draft preparation, Y.L.; writing—review and editing, Y.L., W.L. and M.Z.; funding acquisition, W.L. All authors have read and agreed to the published version of the manuscript.

**Funding:** This research was funded by the National Natural Science Fund of China (Grant No. 41771252, 41901100, 42001035, 41801015, 42101115, 52179026); the National Cryosphere Desert Data Center Program (Grant No. E01Z790208); the Gansu Science and Technology Association Youth Science and Technology Talent Support Project (GXH20210611-09); the Forestry and Grassland Science and Technology Innovation Program of Gansu Province (Grant No. GYCX[2020]01); Key R&D Program of Gansu Province, China (Grant No. 20YF8FA002); the XPCC Science and Technique Foundation (Grant No. 2021AB021); the Think Tank Platform Construction Program of Gansu Association for Science and Technology (Grant No. GSAST-ZKPT[2020]01); the Opening Fund of the National Cryosphere Desert Data Center (No. 2021kf05); and the Opening Fund of the Technology Innovation Center for Mine Geological Environment Rehabilitation Engineering in Alpine and Arid Regions, Ministry of Natural Resources (No. HHGGKK2102).

**Institutional Review Board Statement:** Not applicable.

**Informed Consent Statement:** Not applicable.

**Data Availability Statement:** The data presented in this study are available on request from the corresponding author.

**Acknowledgments:** We are grateful to those who participated in the data processing and manuscript revisions.

**Conflicts of Interest:** The authors declare no conflict of interest.

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
