# Peer review of "Quantitative Assessment for the Spatiotemporal Changes of Ecosystem Services, Tradeoff–Synergy Relationships and Drivers in the Semi-Arid Regions of China"

_remotesensing, doi:10.3390/rs14010239_

Round 1

Reviewer 1 Report

The manuscript implements a quantitative assessment of the spatiotemporal changes of ecosystem services, combined relationships, and collective drivers in semi-arid regions of China. The study results highlighted most of the ecosystem services have a similar spatial distribution in the case study area. The manuscript which involves various predictions and processed maps regarding temporal and spatial hydrogeological responses of the study area is interesting. However, many studies exist which investigate the natural and artificial drivers of environmental variability in the sales of watersheds, large river basins, and regions as well. The manuscript portrays a good quality of scientific and technical writing. The suitability of this article in the journal ‘Remote Sensing’ which primarily focuses on any applications or developments in the technology of remote sensing is clear to me. However, the contribution of the manuscript which makes it stand out is unclear to me. As a reviewer, I feel that the manuscript cannot be published in the present form and needs a strong justification on why this study is unique and deserves publication. I do not recommend accepting this manuscript for publication in the journal, Remote Sensing. 

Certain concerns need to be addressed to improve this manuscript.

  1. Section' 1. Introduction’ is explained well. However, the authors should address the gaps in the existing studies and illustrate how this study bridges those gaps. 

  1. The manuscript entails valuable information. As a reader/reviewer, I feel that it talks specifically about the study area. Rather than tuning this section to the case study area, the authors should highlight the main relevance of this kind of work in the global research community.

  1. Section ‘2. Methods’ involve several equations. I understand that equations 1 to 24 are not developed by the authors. If so, what is the relevance/uniqueness of adopting already existing functions to evaluate ecosystem service components such as carbon storage, water retention, soil retention, sand fixation, food production, and habitat quality?

  1. The authors need to provide a Figure (Flow chart/ Methodology of the work) for the section ‘2. Methods’.

  1. Section ‘3. Results’ is a voluminous one with information and pictorial representations. I would suggest classifying the work as two separate papers, one concerning the spatiotemporal analysis of the six ecosystem components and the other focusing on the hydrological response alteration with the incorporation of these modified variables.

  1. The authors need to check and include the assumptions employed in developing the models of the study. 

  1. Any developed models need to be verified or validated to ascertain their reliability and accuracy. Is there any kind of validation conducted for the developed model/models which predicts the hydrological responses in this study?

  1. A section ‘Limitations of the study’ is recommended.

  1. The reference section needs to be formatted according to the reference formatting style of the journal.

As a reviewer, I do not recommend accepting this manuscript for publication in the journal, Remote Sensing.

Reviewer 2 Report

Dear Sir, Madam

I have had the pleasure of reading your manuscript. I find the research that you performed to be relevant. However I have comments that I would like to share. Specifically

1) I have a hard to understanding why you have submitted the paper to a remote sensing journal. I do see that you have used some remote sensing data (being landsat NDVI). However the impact of this datasource on your methodology/results is not clearly explained. Considering the amount of other datasources that you use, I actually doubt whether the influence is very great. Furthermore, there are remote sensing products for AET, PET, Precipitation, which are not used (even for comparison reasons).

2) I find the paper in general to be very poorly written. I find a lot of spellings errors and even errors in equations that distract greatly from the story that you would like to tell. For example, I believe that the equation at line 270 should be Yx=(1- AET/Px) *Px. As it is written at the moment, the calculation does not make sense.

3) more attention should have been put on why specific ecosystem services were investigated. At present this is only being explained in line 109. However I would have liked to see more information being given on this. For instance, what are all of the ecosystem services in the study area, and why are these so important. Furthermore, are these all of the important ones, or have other studies others to also be important. 

4) I find the methodology section lacking in it's explanation. I would prefer the overall methodology to be explained first (including a thorough explanation regarding the Invest model), followed by the individual methods for calculating services (such as water retention, soil retention), and finally the data sources used for each variable. At this moment, it is not clear to me where which data is being used. Also, I would regard, the paragraph 'statistical analyses' starting on Line 208) to be part of the methodology section.

Specific questions:

L81: Who do you mean by 'researcher' ? Is it a unpublished manuscript. If so, you can still cite this as such).

L217 why was vegetation cover estimated for a different time period than for the rest of NDVI?

L285. If SR represents avoided (!) soil loss, than all the terms on the right should be framed so that 'higher values' of those terms would lead to 'more soil being avoided. However Rainfall erosivity is defined such that higher values indicate higher levels of soil loss. .. This strikes me as odd.

Round 2

Reviewer 1 Report

All the comments are addressed well. As a reviewer, I recommend accepting this manuscript for publication in the journal, Remote Sensing.

Author Response

Dear Reviewers:

Thank you very much, and we further revised again the manuscript to avoid minor mistakes and grammar errors.

Most sincerely,

Dr Yongge Li On Behalf of all Authors

Reviewer 2 Report

Dear Author

Thank you for your speedy revision on your manuscript. I see that you thoroughly modified your paper, leading to a great improvement in the quality of the manuscript. I feel you have taken a lot of care in addressing the comments I provided in the previous version.

However I still would like to see a better explanation on why 'remote sensing' data was used. Below I provide to you some suggestions to allow to do this. 

--

I would add an additional paragraph to highlight the potential of using RS, just after the second to last paragraph that closes of with "There is an urgent need to study the spatiotemporal changes .. well-being in arid inland regions." Specifically, I would introduce a short paragraph 'what the potential of remote sensing to this statement is', 'what the state of the art is' in using remote sensing with the Invest/RWEQ models, and 'whether this has been applied specifically to your study area' (this is missing from the introduction).

To further highlight what the impact of such detailed geospatial remote sensing data is, I would add some lines in your discussion section on this. For this, I would suggest to 'rewrite your 'limitations' paragraph in your discussion section to focus on opportunities instead. For example you presently state "Firstly, given that some of the the relevant data is unavailable and low in accuracy ... ' it might be interesting to identiy which of these non-RS data sources could potentially be replaced by RS proxies (that have a higher spatio-temporal resolution (at least at present)).

Also in regard to your statement on "there are uncertainties in the assessment of ecosystem services", it might be interesting to show a 'sensitivity analysis' (which would also show the impact of the remote sensing variables on the outcomes).

Finally, unrelated to the RS issue, I would move your new Fig3 (which I like a lot) forward to be the first graph in your methodology
